# Endothelial Dysfunction and Neutrophil Degranulation as Central Events in Sepsis Physiopathology

**DOI:** 10.3390/ijms22126272

**Published:** 2021-06-10

**Authors:** Marta Martín-Fernández, Álvaro Tamayo-Velasco, Rocío Aller, Hugo Gonzalo-Benito, Pedro Martínez-Paz, Eduardo Tamayo

**Affiliations:** 1BioCritic, Group for Biomedical Research in Critical Care Medicine, 47003 Valladolid, Spain; mmartin.iecscyl@saludcastillayleon.es (M.M.-F.); rallerf@saludcastillayleon.es (R.A.); hgonzalob@saludcastillayleon.es (H.G.-B.); eduardo.tamayo@uva.es (E.T.); 2Department of Medicine, Dermatology and Toxicology, Faculty of Medicine, Universidad de Valladolid, 47005 Valladolid, Spain; 3Research Unit, Hospital Clínico Universitario de Valladolid, 47003 Valladolid, Spain; 4Haematology and Haemotherapy Service, Hospital Clínico Universitario de Valladolid, 47003 Valladolid, Spain; alvarotv1993@gmail.com; 5Gastroenterology Service, Hospital Clínico Universitario de Valladolid, 47003 Valladolid, Spain; 6Instituto de Ciencias de la Salud de Castilla y León (IECSCYL), 42002 Soria, Spain; 7Department of Surgery, Faculty of Medicine, Universidad de Valladolid, 47005 Valladolid, Spain; 8Anaesthesiology and Critical Care Service, Hospital Clínico Universitario de Valladolid, 47003 Valladolid, Spain

**Keywords:** sepsis, endothelial dysfunction, neutrophil degranulation, biomarkers

## Abstract

Sepsis is a major health problem worldwide. It is a time-dependent disease, with a high rate of morbidity and mortality. In this sense, an early diagnosis is essential to reduce these rates. The progressive increase of both the incidence and prevalence of sepsis has translated into a significant socioeconomic burden for health systems. Currently, it is the leading cause of noncoronary mortality worldwide and represents one of the most prevalent pathologies both in hospital emergency services and in intensive care units. In this article, we review the role of both endothelial dysfunction and neutrophil dysregulation in the physiopathology of this disease. The lack of a key symptom in sepsis makes it difficult to obtain a quick and accurate diagnosis of this condition. Thus, it is essential to have fast and reliable diagnostic tools. In this sense, the use of biomarkers can be a very important alternative when it comes to achieving these goals. Both new biomarkers and treatments related to endothelial dysfunction and neutrophil dysregulation deserve to be further investigated in order to open new venues for the diagnosis, treatment and prognosis of sepsis.

## 1. Introduction

Sepsis is one of the main health care problems worldwide. It has been estimated that 31.5 million cases of sepsis and 19.4 million cases of severe sepsis are treated each year in hospitals around the world, with up to 5.3 million deaths annually throughout the world [1]. Sepsis causes a greater number of deaths than other common diseases, such as breast cancer, prostate cancer or myocardial infarction [2].

More than 1.7 million people in the United States are diagnosed with sepsis each year, which is one every 20 s, and the incidence is increasing every year. Sepsis is the leading cause of death in the United States [3], as 270,000 people die each year from the disease, which is one every 2 min [4]. Up to 87% of cases of sepsis originate in the community [4]. Up to 80% of deaths from sepsis could have been prevented with an early diagnosis and prompt treatment [5]. In addition, survivors of sepsis have a shorter life expectancy and a poorer quality of life [6,7]. Sepsis implies a high hospital cost in Europe, the United States, Asia and South America, with a mean hospital cost per stay of $37,424, $32,421, $13,292 and $24,384, respectively [8,9], doubling the average cost per stay for all other conditions [10].

In Europe, the Sepsis Occurrence in Acutely Ill Patients (SOAP) study elucidated that sepsis could already be the most prevalent disease in intensive care units (ICUs), with a higher mortality and economic cost than other pathologies of similar prevalence [11]. Vincent et al., in a study comparing a decade later the patients of the SOAP study (2002) with those of the Intensive Care Over Nations (ICON) audit (2012), affirmed that the presence of sepsis has slightly increased from 29.6% to 31.9%. However, ICU stays and both hospital mortality and 60-day mortality have remained stable over time [12].

In Spain, the incidence of sepsis has been estimated as 367 cases per 100,000 inhabitants per year, with an associated hospital mortality of 12.8%. Cases of severe sepsis have been established in 104 cases per 100,000 inhabitants per year, with a hospital mortality of 20.7%. In the case of septic shock, an incidence rate of 31 cases per 100,000 inhabitants per year has been estimated, with an associated hospital mortality of 45.7%. In Spain, 17,000 deaths per year due to sepsis have been estimated, of which 70% of patients die during the first 3 days after diagnosis [13]. Sepsis incidences increased 2.7 times, as well as the total costs in the Spanish public health system, from 2008 to 2017 [14], but the incidence and mortality seem to have stabilized in the interval from 2010 to 2013, observing a decrease in hospital stay during this same time interval [15].

In sepsis, endothelial dysfunction and neutrophil dysregulation have been identified as the central events in the physiopathology of sepsis [16,17]. The aim of this review is to explore these scenarios in order to assess new potential biomarkers and treatments for enhancing the diagnosis and prognosis of sepsis.

## 2. Search Strategy and Selection Criteria

The references for this literature review were identified through searches for articles in PubMed, giving priority to those published in the last 10 years. The terms used were “endothelial dysfunction”, “endothelium”, “sepsis”, “neutrophil”, “neutrophil dysregulation”, “immature neutrophils” and “sepsis biomarkers”.

## 3. Endothelium and Sepsis

Endothelial dysfunction is a central event in the physiopathology of sepsis [16]. It plays a crucial role in the pathogenesis of organ failure by enhancing the vascular permeability, fomenting coagulation cascade activation and tissue oedema and compromising the perfusion of vital organs [18].

The vascular endothelium is a semipermeable barrier lining the inner surfaces of blood vessels that controls the exchange of fluids, leucocytes and plasma proteins by coordinately opening and closing the cell junctions that compose it [19]. It prevents the entry of microorganisms into tissues and performs a natural anticoagulant function that prevents the uncontrolled activation of coagulation. A normal vascular endothelium is formed by a layer of endothelial cells on a basement membrane, with the glycocalyx in the luminal region [20].
Glycocalyx: It is an organized layer adhered to a surface matrix that covers the luminal surface of the endothelium, composed by glycoproteins, hyaluronan, sulphated proteoglycans and plasma proteins. It acts as a protective barrier between the blood and vessel wall, helping to regulate leucocyte adhesion, to maintain the endothelial barrier and to inhibit intravascular thrombosis [21].Endothelial cells: They form a continuous layer that covers our vasculature on a basement membrane formed by collagen, nidogens/entactins, laminins and the proteoglycan perlecan. Endothelial cells are linked by tight junctions, adherent junctions and gap junctions [16,22].

### Endothelial Dysfunction in Sepsis

Sepsis produces endothelial dysfunction, forcing a pro-adhesive, procoagulant and antifibrinolytic state in endothelial cells, thus altering the hemostasis, leucocyte trafficking, inflammation, barrier function and microcirculation [23]. During the development of sepsis, the pathophysiological events that affect the integrity of the endothelium are listed below:
Systemic inflammation: A large number of mediators involved in the “molecular storm” that occurs in sepsis initiate and amplify the endothelial damage, such as pathogen-associated molecular patterns (PAMPs), cytokines, bradykinin, the platelet activating factor (PAF), vascular endothelial growth factor (VEGF), fibrin degradation products and reactive oxygen species (ROS) [24,25,26,27]. However, the endothelium is not only passive during sepsis but also stimulates the inflammatory response through the production of chemokines that attract immune cells [27].Glycocalyx degradation and shedding: Glycocalyx shedding occurs as a consequence of the “cocktail” of pro-oxidative and proinflammatory molecules that is generated during sepsis [16,24,28]. This deleterious response is aggravated by the release of components of neutrophil extracellular traps (NETs) and damage-associated molecular patterns (DAMPs), such as glycocalyx degradation products themselves [21,28].Increased leucocyte adhesion and extravasation: The shedding of the glycocalyx exposes the endothelium to leucocyte adhesion [28]. The presence of proinflammatory cytokines during sepsis allows for the adhesion of activated immune cells to the vascular wall and promotes migration to the surrounding tissues by inducing the expression of molecules, such as selectin E (SEL-E), selectin P (SEL-P), intercellular adhesion molecule 1 (ICAM-1) or vascular adhesion molecule 1 (VCAM-1) [24]. During sepsis, disruption of the integrity of the endothelial barrier occurs as a consequence of the adhesion of activated neutrophils [29], which release proteases that contribute to the degradation of binding proteins [20].Destruction of intercellular junctions, disruption of the endothelial barrier and endothelial cell death: The presence of an oxidative and proinflammatory scenario during sepsis induces the disassembly of intercellular junctions, creating spaces between endothelial cells [16,24,28]. Endothelial cell death occurs as a consequence of the release of NETs—specifically, by the action of proteases and cationic proteins [23,30]. The endothelial barrier is disrupted by bacterial toxins, which can directly kill endothelial cells, weakening their cytoskeleton and breaking the intercellular junctions of these endothelial cells [26].Procoagulant and antifibrinolytic state induction: The production of nitric oxide (NO), a potent vasodilator, mediated by inducible nitric oxide synthase (iNOS) is increased in sepsis [24,31]. However, there is a significant reduction in NO production by endothelial synthase nitric oxide (eNOS), which causes a direct alteration of vasodilation and promotes leucocyte and platelet adhesion [25]. The downregulation of the endothelial expression of thrombomodulin and protein C receptors leads to the reduced activation of activated protein C, which plays an anticoagulant function [32]. Endothelial cells release a procoagulant glycoprotein called the tissue factor (TF), while the TF pathway inhibitor synthesis remains inhibited [23]. Platelets and the coagulation cascade activation produce microvascular thrombosis [21]. Furthermore, NETs promote hypercoagulability in patients with sepsis by providing support for the formation of thrombi [23]. Acute vascular dysfunction and leakage contribute to hypotension, local hypoxia, insufficient organ perfusion, ischemia and, ultimately, to organ failure, acute respiratory distress syndrome, shock and death in severe patients [25,33].

## 4. Neutrophils and Sepsis

Leucocytes are responsible for recognizing and eliminating any foreign agent from the body. Therefore, they are a fundamental component both against infection and the development of an inflammatory reaction. We can differentiate five types of leucocytes according to their morphological characteristics: lymphocytes; granulocytes (neutrophils, eosinophils and basophils) and monocytes.

The main functions of granulocytes are phagocytosis and the killing of microorganisms. The origins of granulocytes are in a common progenitor to all blood cells; in a stepwise process of differentiation, proliferation and maturation and in the bone marrow. Totipotent stem cells, under the influence of medullary microenvironment factors, lead to progenitor cells increasingly committed to the myeloid series from which the morphologically recognizable granulocytic precursors originate in the bone marrow [34] (Figure 1).

Neutrophils are the most important cells in the host’s natural defense against microorganisms, especially due to the granules in their cytoplasm:Azurophilic granules. They are lysosomes containing myeloperoxidases and powerful hydrolytic enzymes necessary for the destruction of microorganisms (acid hydrolases; proteases such as proteinase 3, cathepsin G and elastase; cationic proteins such as lysozymes, defensins, azurocidin, bactericidal permeability increasing protein (BPI); etc.) [35].Specific granules. They contain lysozymes; lactoferrin, which has bactericidal and bacteriostatic activity against viruses, fungi and bacteria [36]; lipocalin 2, which also has microbicidal properties; olfactomedin 4; transcobalamin I and other substances involved in the activation of phagocytosis. They are peroxidase-negative.Gelatinase granules. These types of granules are mobilized when neutrophils contact the activated endothelium for the first time. They contain matrix-degrading enzymes, such as gelatinase, and membrane receptors, such as macrophage receptor 1 (MAC-1), CD177 Molecule (CD177), Carcinoembryonic Antigen-Related Cell Adhesion Molecule 8 (CEACAM8), etc., which are essential in the early phases of the inflammatory response of neutrophils and their extravasation into inflamed tissues [37].Secretory vesicles. They are not considered true neutrophil granules, being significantly smaller. They constitute an important reservoir for membrane-associated receptors, such as Matrix Metallopeptidase 25 (MMP25), lymphocyte function-associated antigen-1 (LFA-1) and MAC-1, as well as actin, actin-binding proteins and alkaline phosphatase, which are essential for the establishment of firm contact of the neutrophil with the endothelium-activated vascular system and to complete diapedesis towards inflamed tissues where, through chemotaxis, it locates and eradicates the responsible pathogen [37,38].

Neutrophils basically perform four functions: adhesion, chemotaxis, phagocytosis and bacteriolysis.
Adhesion: Neutrophil migration from blood to tissues is an active process involving a complex set of adhesion molecules on the membrane of the leucocyte that are sequentially activated and have their corresponding receptors on the vascular endothelium. This mechanism allows neutrophils to roll and adhere with progressive firmness to the endothelial surface by selectins, integrins and other molecules and allows their receptors to finally cross the endothelial barrier [34]. Neutrophils are first captured onto the endothelial cell surfaces by the upregulation of adhesive molecules on the endothelial luminal surface in response to inflammatory cytokines and bacteria-derived peptides. Leukocyte selectins mediate these early adhesive interactions, which are transient and weak, promoting the “rolling” of neutrophils on endothelial cells [39]. Upon activation via chemokine receptorsLFA-1, MAC-1 and very late antigen-4 (VLA-4) bind to members of the immunoglobulin superfamily present on endothelial cell membranes, such as intercellular adhesion molecules 1 and 2 (ICAM-1 and ICAM-2) and VCAM-1, respectively [40].Chemotaxis: It is the mechanism by which multiple chemotactic factors (products released by microorganisms, damaged cells, C-X-C Motif Chemokine Ligand 8 (IL-8) and complement fractions) form a chemical gradient that directs the diapedesis of neutrophils to tissues in the precise direction of the focus of infection or inflammation, where they accumulate after passing between the endothelial cells of the microcirculation [34].Phagocytosis: The bacterium or foreign material is recognized and consequently ingested during this process. The membrane then invaginates and simultaneously emits pseudopods, encompassing the particle in a phagosome [34].Bacteriolysis: The formation of the phagosome attracts the granules of the neutrophils, which bind to it, degranulating themselves. The killing of microorganisms occurs, in part, due to the lytic action of the different granular enzymes, but the most important mechanism consists in the generation of oxygen metabolites, with great microbicidal power. Oxygen is reduced by nicotinamide adenine dinucleotide phosphate (NADPH), forming superoxide radicals (O_2_^−^) and generating hydrogen peroxide (H_2_O_2_), which acts as a substrate for myeloperoxidase, which oxidizes halides into hypochlorous acids and chloramines, the latter being powerful microbicides. There is a detoxification mechanism that prevents the excess H_2_O_2_ generated from destroying the granulocytes and damaging the adjacent tissues [34].

Therefore, neutrophils are a fundamental piece in the innate immune response during sepsis by secreting regulatory cytokines, chemokines and leukotrienes; endocytosing pathogens and directly contributing to the destruction of microbes [41]. In sepsis, the existence of neutrophil dysregulation has been documented, leading to an alteration in the directed migration of neutrophils to the site of infection (neutrophil paralysis), where an inadequate antimicrobial response occurs [42]. This impairment of neutrophil recruitment is directly related to mortality in sepsis. During non-severe sepsis, neutrophils expressing C-X-C Motif Chemokine Receptor 2 (CXCR2) are recruited from the blood to the site of infection in response to C-X-C Motif Chemokine Ligand 2 (CXCL2) and other chemoattractant. Neutrophils migrate to the infection site, releasing NETs and producing reactive oxygen and nitrogen intermediates in order to kill pathogens. The systemic spread of pathogens leads to the systemic activation of Toll Like Receptor (TLRs), inducing the expression of tumor necrosis factor-α (TNF-α) and iNOS, which can lead to the upregulation of G Protein-Coupled Receptor Kinase 2 (GRK2), exacerbating the down regulation of CXCR2 on the neutrophil surface, resulting in a failure of migration to the infectious focus [42]. TLR agonists activate endothelial cells and neutrophils, inducing the expression of iNOS, Heme Oxygenase 1 (HO-1) and Peroxisome Proliferator Activated Receptor Gamma (PPARγ), which trigger the reduction of adhesion molecule expression (ICAM-1 on endothelial cells and L-selectin on neutrophils) and CXCR2 desensitization in neutrophils. In addition, an increase in the acute-phase proteins (APPs) serum levels, which may have an additional role in inhibiting neutrophil–endothelium interactions, occurred because of the presence of high levels of circulating proinflammatory cytokines/chemokines [17]. Quantitative neutrophil alterations have also been described in severe forms of sepsis. The presence of low levels of circulating neutrophils in patients with septic shock and fatal outcomes [43] may be due to previous immunosuppressive conditions, a greater adhesion of neutrophils to the vascular endothelium, migration to tissues, increased apoptosis and insufficient production in bone marrow [44].

The production of immature forms of neutrophils could represent a need to expand the circulating neutrophils in response to infection, as well as the replacement of destroyed or consumed neutrophils in the most severe patients. A loss of balance between the mature and immature forms can cause ineffective neutrophil responses during sepsis [41]. An association between the presence of increased immature forms of neutrophils in the blood of septic patients and a poor prognosis has been described [45], as well as an increased risk of mortality after suffering septic shock [46]. These immature forms of neutrophils contain proteases, which contribute to host protection against invading pathogens mediated by the neutrophil oxygen-independent system [47]. These proteases are effective in destroying pathogens, but they can also cause cellular and tissue damage [48,49], as well as cause disruption of the integrity of the endothelial barrier [29], causing immunosuppression in sepsis patients [50]. Moreover, neutrophils play a role in regeneration and repair. This cell is a source of VEGF, which is a key signal protein in blood vessel formation and presents chemotactic effects on endothelial cells [51]. Moreover, neutrophils release other angiogenic factors that can directly activate endothelial growth, such as matrix metalloproteinase 9 (MMP-9) and the cathelicidin antimicrobial peptide (LL-37/hCAP-18) [51]. Matrix metalloproteinases (MMPs) are involved in the cleavage of proteoglycans from the endothelial cell membrane. In this sense, these proteases are activated during inflammatory states by reactive oxygen species (ROS), and proinflammatory cytokines such as TNF-α activate these metalloproteases [52], which participate in vasodilatation and oedema formation and leukocyte adhesion to the epithelium through the expression of adhesion molecules; moreover, it controls blood coagulation, contributes to oxidative stress in sites of inflammation and indirectly induces fever [53]. In this context, the presence of high expression levels of immature neutrophil markers, such as elastase, myeloperoxidase and cathepsin G, has been associated with greater organ failure and mortality [54].

Neutrophil proteases contribute to local neutrophil priming and poor bacterial killing through their adhesion and inactivation of the C5a receptor [55], showing a correlation between the low levels of C5a receptors on neutrophils and disease severity [56]. The contribution of neutrophil proteases in the physiopathology of disseminated intravascular coagulation (DIC) in sepsis has also been described, promoting, together with externalized nucleosomes, the formation of thrombi in blood vessels [57].

## 5. Sepsis Biomarkers

The diagnosis of sepsis is still complicated due to the absence of a key symptom to detect it. This can lead, on the one hand, to the nonidentification of patients, with a consequent delay in treatment and a higher mortality, and on the other hand, the overdiagnosis of patients, with a consequent overtreatment with antibiotics [58].

In sepsis, an early diagnosis and rapid decision-making are essential for a prompt implementation of the corresponding treatments, which are vital for the prognosis and survival of patients. Therefore, it is essential to have fast and reliable diagnostic tools. In this regard, the use of biomarkers could be a feasible alternative for achieving these goals. In this sense, transcriptomics has proven to be a very useful discovery tool for improving the diagnosis and prognosis of sepsis [54,59,60,61,62].

On the other hand, proteomics are presented as a fundamental instrument in biomedical research, both to develop diagnostic and prognostic tools and to search for new therapeutic targets for the design of drugs and vaccines. The evaluation of biomarkers of a protein nature is technically simpler than for those of transcriptomic origins. The development of “point-of-care” devices can contribute to the implementation of biomarkers in the clinical routine [63].

A large number of mediators and molecules are released and interact during the pathophysiological process of sepsis, which can be useful as both diagnostic and prognostic markers of this disease. So far, more than 180 biomarkers have been studied in sepsis, which is reflected in the large number of studies in this regard [64,65,66].

In this pathology, a series of classic biomarkers have been widely used for the diagnosis of infection and to evaluate the prognosis of the disease: C-reactive protein (CRP) and procalcitonin (PCT).

CRP is a glycosylated protein, an acute-phase reactant, that is synthesized by liver hepatocytes stimulated by IL-6 and IL-8 in response to tissue damage or infection [67,68].

The production of this protein is increased 4–6 h after the initial insult, being able to rise in the first 24–48 h several hundred times above its basal level, remaining elevated during the acute phase of the infection and even letting the aggression cease [69]. The cut-off point for this protein established for detecting infections was set at around 10 mg/dL, with a sensitivity of 88% and a specificity of 58%, observing a correlation between the plasma concentration and severity of infection [70]. The modest specificity of CRP makes it a biomarker with certain limitations, sometimes causing false positives derived from noninfectious inflammatory processes [71]. Furthermore, it presents slow-elimination kinetics, showing a lower prognostic value than that of the other biomarkers [72].

PCT is a precursor of the calcitonin hormone that is physiologically synthesized by thyroid C cells [73]. Its concentration rises rapidly in the blood during a severe bacterial infection, such as sepsis, and it can be detected in the first 2–4 h after insult, reaching a maximum concentration between 24 and 48 h [74]. Thus, it could promote an early detection of infection, as well as monitor the evolution and response to treatment [75].

An elevated PCT level is not exclusive to bacterial infections, since it can also be elevated in other scenarios, such as systemic fungal infections, trauma or in the pathologies related to kidney failure [76]. Despite this, PCT presents a greater sensitivity and specificity for the detection of sepsis compared to the other biomarkers [77].

The basal blood values of PCT are below 0.5 ng/mL. Values above this reference can occur not only in patients with infection but, also, in other pathologies, such as autoimmune diseases, trauma, cardiac processes, surgeries, burns and pancreatitis [78]. The presentation of PCT values above 10 ng/mL is associated with a higher probability of severe sepsis and septic shock [79]. In addition, PCT could help in guiding antibiotic treatment and de-escalation [80].

Biomarkers other than CRP and PCT have been evaluated for their potential use in sepsis.

Lactate, which is currently included as one of the parameters to define the presence of septic shock in the SEPSIS-3 criteria [81,82], has a good capacity to assess the evolution and prognosis of patients. Increased anaerobic metabolism during sepsis leads to the onset of hypoxemia and tissue hypoperfusion, which is reflected in the elevation of the lactate levels. The presence of elevated lactate levels is strongly related to a higher hospital mortality [83].

Cytokines represent another type of biomarker associated with sepsis. Cytokines are regulators of the immune response playing an essential role in regulating inflammation and trauma. Proinflammatory cytokines stimulate systematic inflammation, while anti-inflammatory cytokines inhibit inflammation and enhance healing. Among the major proinflammatory cytokines that induce early responses are interleukin-1α (IL-1α), IL-1β, IL-6 and TNF-α. Anti-inflammatory cytokines are involved in the prevention of potentially harmful effects of persistent or excess inflammatory reactions. The major anti-inflammatory cytokines include the IL-1 receptor antagonist (IL-1Ra), IL-4, IL-6, IL-10, IL-11 and IL-13 [84]. There is a concomitant presence of high levels of proinflammatory cytokines, such as IL-6, IL-8, TNF-α and monocyte chemoattractant protein-1 (MCP-1), and anti-inflammatory cytokines, such as IL-10 and IL-1RA, at the time of diagnosis in severe forms of sepsis [41]. It has also been described that the presence of both proinflammatory and anti-inflammatory cytokines, combined in the form of scores, in patients with sepsis are associated with a worse prognosis [85].

Proadrenomedullin (proADM) has also been studied for its potential utility as a biomarker in sepsis. Adrenomedullin regulates the vascular tone and endothelial permeability [86], with its intermediate part, proADM, being used as a biomarker due to its greater stability. Its levels are elevated in cardiac, respiratory, renal and sepsis pathologies, with its basal concentration in normal conditions being minimal. Its production is increased as a consequence of oxidative stress, proinflammatory cytokines such as TNF-α and IL-1, hormones such as angiotensin II and aldosterone and hypoxia factors or hyperglycemia [87]. ProADM increases in septic shock because of the decreased clearance and increased cytokine production [88]. MR-proADM is the middle fragment of proADM that is more stable and directly reflects the level of the active peptide (adrenomedullin), which is rapidly degraded [89]. MR-proADM has been described as a potential marker of disease progression in patients with a suspected infection in the emergency department [90], as well as a predictor of organ failure in patients with community-acquired pneumonia [91]. Therefore, it could represent a diagnostic and prognostic biomarker in sepsis, since the presence of high levels is associated with a greater severity and mortality.

Despite all the biomarkers evaluated for sepsis, no single biomarker presenting sufficient sensitivity and specificity for clinical use has been established at present. In the context of sepsis, a biomarker should be able to reflect the presence of an infection, as well as its evolution and response to treatment [92,93].

Based on the important role of endothelial dysfunction and neutrophil degranulation in the physiopathology of sepsis, it seems plausible that monitoring these two pathophysiological events could be helpful in predicting or detecting sepsis, as well as to evaluate its evolution and prognosis.

Numerous biomarkers of endothelial dysfunction have been evaluated [94,95,96,97], including the most relevant markers of endothelial cell activation (angiopoietin 2 [96] and endocan [98]); glycocalyx degradation markers (sindecan 1, hyaluronic acid, heparan sulphate and chondroitin sulphate [99]); vasoactive peptides (MR-ProADM [100]); cell adhesion molecules (selectins [96], ICAM-1 and VCAM-1 [98]); coagulation inhibitors (THBD) [95] and molecules with vasoconstrictor and vasopressor activities (endothelin [96]), among others.

In the context of neutrophil degranulation, there are numerous studies that revealed the potential role of neutrophil proteases in the diagnoses and prognoses of sepsis [54,89]. Matrix Metallopeptidase 8 has been identified as a marker to distinguish sepsis [101,102] and estimate the probability of mortality in septic shock [103], as well as a novel modulator of inflammation during sepsis and a potential therapeutic target in this disease [104]. Olfactomedin 4 has been proposed as a candidate marker of a pathogenic neutrophil subtype in septic shock [105]. Matrix Metallopeptidase 8 and Olfactomedin 4 have been associated with sepsis-induced respiratory distress [106]. PRTN3 is involved in the existence of greater endothelial permeability [107]. Lipocalin 2 has been shown to be useful for the early diagnosis, risk stratification and prognosis of sepsis in the emergency department [108,109], as well as for the detection of multiorgan dysfunction and mortality in patients with sepsis and septic shock [110,111]. Myeloperoxidase and Lactoferrin have also been described as characteristic hallmarks of sepsis [112].

Therefore, a multimarker strategy combining the biomarkers of endothelial damage and neutrophil degranulation could be very useful for the diagnosis and prognosis of sepsis, as well as for monitoring its evolution [89].

## 6. Clinical Practice Implications

Sepsis is a time-dependent disease, and for this reason, it is important to get an immediate and specific antibiotic treatment for this condition. In this sense, it has been reported that the case fatality rate increases by 20% for each one-hour increase in the door-to-antimicrobial time in septic patients [113]. Moreover, the delay of antibiotic treatment initiation is associated with an increase in the death rate [114].

There are progressively more and more studies showing the importance of endothelial damage, emergency granulopoiesis and neutrophil degranulation in the development of sepsis and the potential role of evaluating biomarkers that monitor these aspects in the assessment of a patient [89].

This opens the door to the incorporation of new tools that, through fast, reliable and reproducible technologies, allow the evaluation of the entire spectrum of the disease, as well as new treatment opportunities.

An example would be the combination of the antibiotic treatment with drugs that protect the endothelium in patients with infections, which could prevent the development of sepsis or improve its evolution. There are already numerous studies that propose therapeutic options to prevent or treat endothelial dysfunction [97,115,116,117], of which one of the most promising today is that by Marik et al., which showed that the administration of vitamin C, corticosteroids and thiamine prevented the progression of organ dysfunction and reduced the mortality in patients with sepsis [118]. In another work, it was also demonstrated that hydrocortisone and ascorbic acid exerted a synergistic effect, preventing and repairing the dysfunction of the endothelial barrier induced by lipopolysaccharide [119] (Figure 2).

Another example would be to focus on treatments oriented towards the roles played by the proteins contained in neutrophil granules. The true roles these proteins play in sepsis remain unknown to date. On the one hand, there is evidence of the deleterious role of these proteins on the vascular endothelium by inducing changes in the transendothelial permeability, neutrophil and platelet adhesion and the loss of its integrity [120,121,122]. On the other hand, the elevation of these proteins could suppose a reactive mechanism with the aim of protecting the endothelium [123] or preventing the deleterious effects caused by NETs, which have a protective function against pathogens but are also related to the development of thrombosis and excessive inflammation [124]. Therefore, a possible strategy could be the downmodulation or blocking of the neutrophil granule protein expression, which could prevent or reduce the endothelial dysfunction and provide a clinical benefit in sepsis [125,126] (Figure 2). There is a growing arsenal of emerging drugs that can inhibit the effects caused by the proteins contained in neutrophil granules [127].

**Figure 2 ijms-22-06272-f002:**
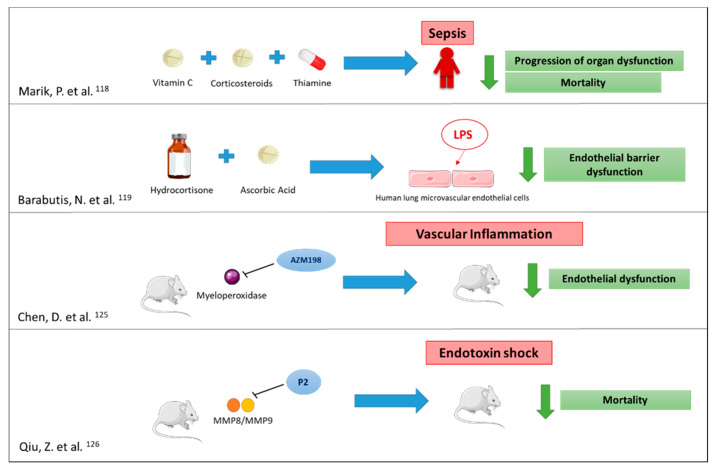
Example of the potential treatments to prevent/treat endothelial damage and to downmodulate/block the neutrophil granule protein expression in order to provide a clinical benefit for sepsis. Figures were obtained from https://smart.servier.com, accessed on 5 May 2021.

## 7. Conclusions

Endothelial dysfunction and neutrophil dysregulation are present in the context of severe infections. There is growing evidence supporting the potential value of the molecules involved in endothelial dysfunction and neutrophil degranulation for the diagnosis and prognosis of sepsis. Further studies should elucidate if the monitoring of endothelial dysfunction and neutrophil function could help to identify early sepsis and, also, whether the administration of drugs to control these events could provide a beneficial effect in the clinical management, prevention or prognosis of sepsis.

## Figures and Tables

**Figure 1 ijms-22-06272-f001:**
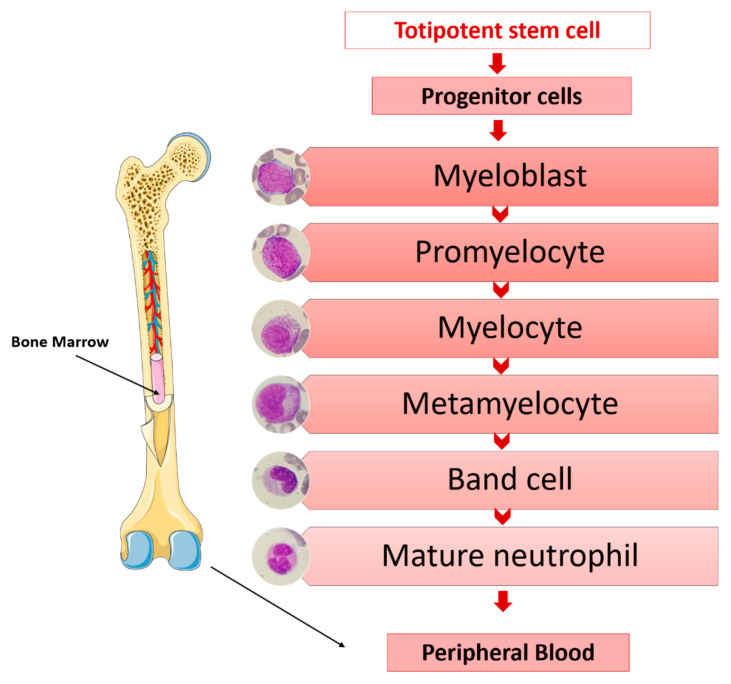
Sequence of granulocyte maturation in the bone marrow.

## Data Availability

Not applicable.

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
