# Peer review of "Endothelial Dysfunction and Neutrophil Degranulation as Central Events in Sepsis Physiopathology"

_ijms, 2021, doi:10.3390/ijms22126272_

Round 1
Reviewer 1 Report
Overall this review of the literature is well written and clearly presented. The authors did a thorough job of capturing the literature including identifying opportunities for intervention of neutrophil:endothelial damage.
Line 179: CD11b/CD18 and MAC1 are the same thing. I don’t think MAC1 is in secretory granules, please double check.
The authors should consider the fact that the single biggest barrier regarding controlling neutrophil damage is that it is hard to do that without causing immunosuppression in a septic patient. Wolf et al made progress in that regard:
NATURE COMMUNICATIONS | (2018) 9:525 | DOI: 10.1038/s41467-018-02896-8 | www.nature.com/naturecommunications
Author Response
We greatly appreciate all the suggestions made by Reviewer 1 to improve the manuscript. All the specific suggestions of the reviewer have been incorporated into the new version, as detailed below.
Overall this review of the literature is well written and clearly presented. The authors did a thorough job of capturing the literature including identifying opportunities for intervention of neutrophil:endothelial damage.
-Line 179: CD11b/CD18 and MAC1 are the same thing. I don’t think MAC1 is in secretory granules, please double check.
Attending to the reviewer's concern, we have changed CD11b/CD18 by MAC-1 in the manuscript. Regarding the role of MAC-1 in the secretory granules, it has been checked in the references Cowland and Borregaard (2016) and Rørvig et al., (2013). Moreover, we have found other reference about this question. In this sense, Sengeløv et al., (1993) reported that 75% of MAC-1 colocalized with specific granules including gelatinase granules, 20% with secretory vesicles and the rest with plasma membranes in unstimulated neutrophils.
Cowland, J.B.; Borregaard, N. Granulopoiesis and Granules of Human Neutrophils. Immunol. Rev. 2016, 273, 11–28.
Rørvig, S.; Østergaard, O.; Heegaard, N.H.H.; Borregaard, N. Proteome Profiling of Human Neutrophil Granule Subsets, Secretory Vesicles, and Cell Membrane: Correlation with Transcriptome Profiling of Neutrophil Precursors. J. Leukoc. Biol. 2013, 94, 711–721.
Sengeløv, H.; Kjeldsen, L.; Diamond, MS.; Springer, T.A.; Borregaard, N. Subcellular localization and dynamics of Mac-1 (alpha m beta 2) in human neutrophils. J. Clin. Invest. 1993, 92, 1467–1476.
-The authors should consider the fact that the single biggest barrier regarding controlling neutrophil damage is that it is hard to do that without causing immunosuppression in a septic patient. Wolf et al made progress in that regard: NATURE COMMUNICATIONS | (2018) 9:525 | DOI: 10.1038/s41467-018-02896-8 | www.nature.com/naturecommunications
As suggested by the reviewer, we have considered the control of neutrophil damage without causing immunosuppression. In this sense, we have added the reference Wolf et al. (2018) in the section “4. Neutrophils and sepsis”.
Wolf D; Anto-Michel N; Blankenbach H; Wiedemann A; Buscher K; Hohmann JD; Lim B; Bäuml M; Marki A; Mauler M; Duerschmied D; Fan Z; Winkels H; Sidler D; Diehl P; Zajonc DM; Hilgendorf I; Stachon P; Marchini T; Willecke F; Schell M; Sommer B; von Zur Muhlen C; Reinöhl J; Gerhardt T; Plow EF; Yakubenko V; Libby P; Bode C; Ley K; Peter K; Zirlik A. A ligand-specific blockade of the integrin Mac-1 selectively targets pathologic inflammation while maintaining protective host-defense. Nat. Commun. 2018, 9, 525.

Reviewer 2 Report
The authors in this review have discussed about the endothelial dysfunction and neutrophil degranulation as central events in the sepsis physiopathology. Overall, the manuscript is well written, however, several other manuscripts have been published previously in various peer reviewed journals highlighting the importance of these events in sepsis. Following are some major comments.
- Considering the dynamics in sepsis field it will be relevant if authors can provide the time of search in the “Search strategy and selection criteria”.
- What is the rationale of citing data from 2010 to 2013 for the mortality rate of sepsis in the Spanish public health system? Author should cite contemporary stats and if possible pertinent to the COVID19 impact.
- The regulation of TNF alpha (i.e. its shedding, metalloprotease activation) and its implication in sepsis should be discuss in more detail.
- The process of neutrophil paralysis (i.e. impairment of neutrophil recruitment to the cite of infections) should be discuss in more detail.
- Neutrophils also play role in regeneration and repair and author should discuss.
- The adhesion molecules and their regulations should be describe in more detail.
- Role of antibiotics in the sepsis treatment and there role on neutrophil pathophysiology should be discuss.
Author Response
We would like to thank reviewer for the comments and suggestions on our manuscript. We acknowledge most of the criticism and we have made an effort to address conveniently the concerns and suggestions, which greatly improved the manuscript. We hope that the revised version is now according to the reviewer standards. Here is our detailed reply to reviewer.
The authors in this review have discussed about the endothelial dysfunction and neutrophil degranulation as central events in the sepsis physiopathology. Overall, the manuscript is well written, however, several other manuscripts have been published previously in various peer reviewed journals highlighting the importance of these events in sepsis. Following are some major comments.
- Considering the dynamics in sepsis field it will be relevant if authors can provide the time of search in the “Search strategy and selection criteria”.
We understand the reviewer concerns about this point. However, in “2. Search strategy and selection criteria” section we have stated the time of search: “References for this literature review were identified through searches for articles in PubMed, giving priority to those published in the last 10 years”.
- What is the rationale of citing data from 2010 to 2013 for the mortality rate of sepsis in the Spanish public health system? Author should cite contemporary stats and if possible pertinent to the COVID19 impact.
Following the reviewer’s suggestion, we have added the reference Darbà and Marsà (2019), where the data of sepsis in Spain is up to date.
Darbà, J.; Marsà, A. Epidemiology, management and costs of sepsis in Spain (2008-2017): a retrospective multicentre study. Curr. Med. Res. Opin. 2020, 36, 1089–1095.
Regarding the question about the COVID-19 impact, maybe SARS-CoV-2 could be affect the sepsis incidence, but we think that this is not the goal of this review and it could be distracting to readers. However, we think that the reviewer's question is a good idea for a future review.
-The regulation of TNF alpha (i.e. its shedding, metalloprotease activation) and its implication in sepsis should be discuss in more detail.
We greatly appreciate the question made by the reviewer. In this sense, we have included more information about the regulation of TNF-α in “4. Neutrophils and sepsis” and “5. Sepsis biomarkers” sections. In addition, we have reviewed other pro-inflammatory cytokines that could be of interest for the readers.
In this sense, we have added the references listed below:
Uchimido, R.; Schmidt, E.P.; Shapiro, N.I. The Glycocalyx: A Novel Diagnostic and Therapeutic Target in Sepsis. Crit. Care 2019, 23, 16.
Zelová, H.; Hošek, J. TNF-α Signalling and Inflammation: Interactions between Old Acquaintances. Inflamm. Res. 2013, 62, 641–651.
- Neutrophils also play role in regeneration and repair and author should discuss.
As suggested, we have added more information about the play of neutrophils in regeneration and repair of endothelium. So, we have included the reference Wang, J. Neutrophils in tissue injury and repair. Cell Tissue Res. 2018, 371, 531–539.
-The adhesion molecules and their regulations should be describe in more detail.
Following the indications of the reviewer, we have included more information about the adhesion molecules of neutrophils in “4. Neutrophils and sepsis” section. For this reason, we have added the below references:
Filippi, M.-D. Neutrophil Transendothelial Migration: Updates and New Perspectives. Blood 2019, 133, 2149–2158.
Muller, W.A. Getting Leukocytes to the Site of Inflammation. Vet. Pathol. 2013, 50, 7–22.
- Role of antibiotics in the sepsis treatment and there role on neutrophil pathophysiology should be discuss.
Following the reviewer's suggestion, we have included two references about the importance of the immediate and specific antibiotic treatment (Peltan et al., 2017; Peltan eta l., 2019).
Peltan, I.D.; Mitchell, K.H.; Rudd, K.E.; Mann, B.A.; Carlbom, D.J.; Hough, C.L.; Rea, T.D.; Brown, S.M. Physician variation in time to antimicrobial treatment for septic patients presenting to the emergency department. Crit. Care Med. 2017, 45, 1011–1018.
Peltan, I.D.; Brown, S.M.; Bledsoe, J.R.; Sorensen, J.; Samore, M.H.; Allen, T.L.; Hough, C.L. ED Door-to-antibiotic time and long-term mortality in sepsis. Chest 2019, 155, 938‒946.
Regarding the last question, we have only found one reference about the role of antibiotics on neutrophil pathophysiology. In mice treated with antibiotics, the surface expression of the chemokine receptor CXCR2 on neutrophils was reduced, what can explain the inability to move to the infection location. Nevertheless, we honestly think that maybe this information is not relevant for this section and could be discussed in future reviews.
Watanabe, K.; Gilchrist, C.A.; Uddin, J.; Burgess, S.L.; Abhyankar, M.M.; Moonah, S.N.; Noor, Z.; Donowitz, J.R.; Schneider, B.N.; Arju, T.; Ahmed, E.; Kabir, M.; Alam, M.; Haque, R.; Pramoonjago, P.; Mehrad, B.; Petri Jr, W.A. Microbiome-mediated neutrophil recruitment via CXCR2 and protection from amebic colitis. PLoS Pathog. 2017, 13, e1006513.
Round 2
Reviewer 2 Report
The authors have tried their best to address all the previously raised concerns and I have no further comments.